# Failures in Reflective Functioning and Reported Symptoms of Anxiety and Depression in Bereaved Individuals: A Study on a Sample of Family Caregivers of Palliative Care Patients

**DOI:** 10.3390/ijerph191911930

**Published:** 2022-09-21

**Authors:** Vittorio Lenzo, Alberto Sardella, Alessandro Musetti, Maria Cristina Petralia, Irene Grado, Maria C. Quattropani

**Affiliations:** 1Department of Social and Educational Sciences of the Mediterranean Area, University for Foreigners “Dante Alighieri” of Reggio Calabria, 89125 Reggio Calabria, Italy; 2Sisifo-Consortium of Social Cooperatives, 95125 Catania, Italy; 3Department of Clinical and Experimental Medicine, University of Messina, 98125 Messina, Italy; 4Department of Humanities, Social Sciences and Cultural Industries, University of Parma, 43121 Parma, Italy; 5Department of Educational Sciences, University of Catania, 95124 Catania, Italy

**Keywords:** health psychology, palliative care, caregiver, cancer, anxiety, depression, mentalizing

## Abstract

*Introduction.* This study aims at examining the role of failures in reflective functioning in predicting anxiety and depression among family caregivers of palliative care patients deceased for at least one year. *Methods.* A sample of 157 bereaved participants (77.1% females, mean age = 43.50 ± 14.04 years) completed the Hospital Anxiety and Depression Scale (HADS) and the Reflective Functioning Questionnaire (RFQ). *Results.* Results of the correlational analysis showed that anxiety was positively correlated with uncertainty about mental states, indicating one type of impairment in reflective functioning. Anxiety was also negatively correlated with the certainty about mental states. Depression was negatively correlated with certainty but not with uncertainty about mental states. The results of regression analysis indicated that gender and certainty about mental states were statistically significant predictors of anxiety, with the final model explaining 23% of the variance. The results also showed that gender, the condition of being the main caregiver, and the certainty about mental states were significant predictors of depression, with the final model predicting 14% of the variance. *Conclusions.* Overall, the results of this study point out that the bereaved individuals who scored low on certainty about mental states reported more symptoms of anxiety and depression. Psychological interventions to prevent mental disorders and to promote psychological health in the context of palliative care should carefully consider these findings.

## 1. Introduction

The death of a loved one such as a spouse or a close family member represents one of the most stressful events that people face throughout their lives [1]. Besides immunological and endocrine changes, reactions to loss include cognitive, behavioural, physiological–somatic, and affective manifestations [2]. The reported symptoms of depression and distress, as well as anxiety and fears, characterize the former and cause significant impairment in quality of life and constitute distinct phenomena from Prolonged Grief Disorder [3]. Although grief is an inevitable experience of life, it was well established that reaction to the loss may be variegate among individuals [4]. However, bereavement research struggles with several methodological limits, and thereby many important questions on the factors underneath variance in grief outcome remain unanswered [5]. Such an issue assumes fundamental importance in the context of palliative care on the ground that family caregivers who assist their loved ones face a stressful experience that may lead to negative consequences for mental health even after the loss [6].

The burgeoning recent literature does not focus exclusively on prolonged grief disorder. As one way to better grasp the variations in the grief outcome between individuals, several researchers recently zeroed in on the role of psychological factors in the onset of prolonged grief disorder. The focus on psychological factors associated with maladaptive grief reactions was bolstered by the intent of preventing mental disorders as well as the onset of symptoms of depression and anxiety. In this regard, attachment theory has been deemed as a crucial tool for understanding the grief reaction insofar as the loss of a loved one may potentially have negative effects on the attachment system of the mourner [7]. Since that time, an increasing number of studies have depicted a relationship between insecure attachment (i.e., avoidance and anxious factors) and grief reactions [8]. Despite the significant heterogeneity of the research in this field, studies converged in indicating that only 9.8 percent of the bereaved show maladaptive reactions, even though reported symptoms of anxiety and depression are more common [9]. Turning to the psychological factors underneath symptoms of psychopathology, a growing number of studies have examined the role of mentalizing, operationalized as reflective functioning, in a wide array of mental disorders [10,11]. Mentalizing or reflective functioning regards the capacity to understand oneself and others in terms of mental states, such as feelings, beliefs, attitudes, and yearning [12]. Mentalizing capacities emerge within the development of the attachment system, even though they remain two distinct constructs [13,14]. It turned out that failures in mentalizing contribute to explaining pathways to depression and anxiety. For example, Luyten and colleagues argued that mentalizing impairment rooted in insecure attachment relationships and more generally in adverse early childhood experiences may lead to major depressive disorder [15]. Despite mentalizing representing a transdiagnostic and transtheoretical construct for understanding vulnerability to psychopathology and the onset of mental disorders [16], no research so far has investigated the relative contribution of this construct to symptoms reported by family caregivers who lost a loved one. To date, the few studies carried out in the context of loss have shown that mentalizing deficits are related to major depressive disorder and complicated grief [17,18]. In this vein, in order to intervene in time to prevent the onset of depressive disorder and/or anxiety disorder, it may be useful to investigate the relationship between these last and mentalizing in individuals who do not have an established diagnosis of mental disorder, even though they struggle with the consequences of the loss of a loved one. Thus, in pursuing this question, more research is still needed.

In line with these premises, the first aim of this study was to explore the relationships between mentalizing impairments, reported symptoms of anxiety, and depression in a sample of family caregivers of patients who were assisted in palliative home care. We hypothesized that positive relationships would be found between mentalizing impairments, anxiety, and depression insofar as the former is a transdiagnostic concept involved in a broad range of psychological problems. The second aim of this study was to examine the role of reflective functioning in predicting the reported symptoms of depression and anxiety. We hypothesized a significant contribution of reflective functioning in predicting anxiety and depression, even after controlling for sociodemographic and loss-related variables.

## 2. Materials and Methods

### 2.1. Participants

The present study is part of a research project named “Risk and protective factors for prolonged grief disorder in family caregivers of patients in palliative home care”. A priori power analysis [19], conducted using G*Power (version 3.1.9.4) [20], ensured that the statistical power was adequate to obtain statistically significant results. Statistical power was calculated as a function of required power and the significance levels. We selected the F test and linear multiple regression with a fixed model and *R^2^* increase. The statistical power of 0.95 with a total sample size of 146 subjects was obtained by entering a significant finding (at the 0.05 level), a medium effect size (Cohen’s d = 0.15), and a total number of 6 predictors.

### 2.2. Procedure

The research was conducted in accordance with the 1964 Declaration of Helsinki and its later amendments. Privacy of the participants was guaranteed in accordance with the European Union General Data Protection Regulation 2016/679. This study was approved by the Research Ethics Committee for Psychological Research of the University of Messina (n. 93120). Participants signed informed consent forms regarding publishing their data anonymously. The instruments were administered in the context of three palliative home care services by two clinical psychologists between 7 February 2020 and 9 November 2021, and the administration took about 30 min to complete. A total of 159 subjects consented to participate and 157 completed the protocol.

### 2.3. Measures

The participants completed a questionnaire including single-item questions on socio-demographic variables as well as years of bereavement, type of relationship with the deceased, and condition of being the main caregiver. Moreover, the Italian versions of the following self-report instruments were administered:

The Hospital Anxiety and Depression Scale (HADS) [21,22] measures anxiety and depression at a given point in time. The HADS consists of 14 items on a 4-point Likert scale, ranging from “0” to “3”, and is divided into two subscales, each of which includes 7 items. These two subscales provide a measure of anxiety (HADS-A) and depression (HADS-D) by summing the score of 7 items belonging to the subscale. Consequently, scores for each subscale range from “0” to “21” with high scores denoting higher levels of anxiety and depression.

The Reflective Functioning Questionnaire (RFQ) [23,24] consists of 8 items and it evaluates the ability to interpret both the self and others in terms of internal mental states, also named “mentalizing”. More specifically, the RFQ provides a measure of two broad types of mentalizing impairments, which are uncertainty (RFQ U) and certainty (RFQ C) about the mental states of self and others. The items are rated on a 7-point Likert scale ranging from “1” (completely disagree) to “7” (completely agree). Concerning RFQ U, individuals are asked to answer to items such as “Sometimes I do things without really knowing why” or “I always know what I feel.” and then the responses are recoded from 0 to 3. Concerning RFQ C, individuals are asked to answer to items such as “People’s thoughts are a mystery to me” or “When I get angry I say things without really knowing why I am” and the responses are recoded from 3 to 0. Higher scores on RFQ U suggest hypomentalizing (i.e., concrete thinking and poor understanding of the mental states), while high scores on RFQ C reflect hypermentalizing, which is an over-mentalizing disposition not supported by the evidence and, consequently, not grounded on reality. Unlike these two types of impairments, a certain degree of disagreement indicates acknowledgment of the opaqueness of one’s own and others’ mental states, indicating adaptive mentalizing.

### 2.4. Statistical Analysis

Statistical analysis was performed using IBM SPSS Statistics version 26 (IBM Corporation, Armonk, NY, USA). Data obtained from this study were checked; then descriptive and inferential statistical analyses were carried out. Relationships between RFQ and HADS were performed with Pearson product-moment correlation coefficients. Moreover, to examine the relationship between HADS and RFQ, two hierarchical regression analyses were conducted. Specifically, two hierarchical regressions were performed to determine whether the two types of failures in reflective functioning (Uncertainty and Certainty about mental states) significantly improved the proportion of explained variance in the dependent variables. HADS-A was set as the dependent variable in the first regression, while HADS-D was set as the dependent variable in the second one. Age and gender were added as covariates in all three steps. In the second step, we added the year of bereavement and if the bereaved was the main caregiver. In the third step, we added the two factors of mentalizing, which are Uncertainty (RFQ U) and Certainty (RFQ C) about mental states.

## 3. Results

### 3.1. Demographic and Loss Characteristics of the Sample

Table 1 shows the demographic and loss characteristics of the sample. The sample consisted of 157 subjects (36 males and 121 females) between 18 and 81 years (M = 43.50 ± 14.04). Most of them were female (77.1%) and held a high school diploma (42%). The time since the loss was on average 3.59 years (SD = 4.92, range = 1–28) and 52.3% of the bereaved were sons or daughters, while 57.3% were the main caregiver during the illness. Subjects who were the main caregivers showed higher scores of depression (M = 7.18 (SD = 3.66) vs. M = 4.94 (SD = 3.01), *t*(155) = 4.08, *p* < 0.0001), while scores of anxiety did not differ (M = 7.18 (SD = 4.18) vs. M = 7.57 (SD = 4.03), *t*(155) = 4.08, *p* = 0.559).

### 3.2. Descriptive and Correlational Analyses between Anxiety, Depression, and Reflective Functioning

Table 2 shows the descriptive statistics and results of correlation analyses. Results showed that HADS-A positively correlated with RFQ U (r = 0.17; *p* < 0.01) and negatively correlated with RFQ C (r = −0.30; *p* < 0.05). HADS-D was negatively correlated with RFQ C (r = −0.21; *p* < 0.01) but not with RFQ U. Additionally, HADS-A was positively correlated with HADS-D (r = 0.61; *p* < 0.01), while RFQ C was negatively correlated with RFQ U (r = −0.32; *p* < 0.01). Lastly, the time since the loss was negatively correlated with HADS-D (r = −0.16; *p* < 0.05).

### 3.3. Regression Analyses for Anxiety

Table 3 shows the regression results of the effects of reflective functioning (i.e., RFQ U and RFQ C) on HADS-A, controlling for age and gender. In explaining anxiety, gender (β = 0.34; *p* < 0.01) was statistically significant in step 1. This effect persisted in both steps 2 and 3. After adding reflective functioning factors, RFQ C (β = −0.29; *p* < 0.01) was statistically significant to explain HADS-A scores. No significant effect was found for years from the loss and the condition of being the main caregiver. The final model was statistically significant with R^2^ reaching 0.23.

### 3.4. Regression Analyses for Depression

Table 4 shows the regression of the effects of reflective functioning on HADS-D. The model was controlled for age and gender. In the first step, gender (β = 0.20; *p* < 0.05) was statistically significant in explaining depression. This effect did not persist in step 2, whereas the condition of being the main caregiver (β = −0.22; *p* < 0.05) reached statistical significance. Lastly, in step 3, gender (β = 0.20; *p* < 0.05), the condition of being the main caregiver (β = −0.23; *p* < 0.05), and RFQ C (β = −0.17; *p* = 0.01) were all statistically significant with *R*^2^ reaching 0.14.

## 4. Discussion

This study aimed at investigating the role of mentalizing impairments in predicting the reported symptoms of anxiety and depression among bereaved caregivers of patients who were assisted in palliative home care. Although research examining the interplay between mentalizing and psychopathology is certainly impressive, the lack of study in the context of palliative care stands out.

First of all, we sought to explore the associations between reflective functioning, anxiety, and depression. As expected, we found that the scores of anxiety and depression were positively and highly correlated. In this regard, a meta-analysis pointed out that widows show a high prevalence of symptoms of depressive and anxiety disorders [25]. Interestingly, the occurrence of depressive symptoms seems to be independent of age and sex, and thereby it may hinge on the role of certain psychological predictors, such as reflective functioning. Consistently with other studies, our results revealed that hypomentalizing was negatively associated with hypermentalizing, highlighting their opposed meaning [10,11,16,26]. However, because the structure of RFQ includes four shared items between the two subscales, this is not a surprising finding.

The second aim of this study was to explore the role of reflective functioning, together with sociodemographic and loss-related variables, in predicting the reported symptoms of anxiety and depression in bereaved caregivers. Failures or imbalances in reflective functioning have been found in many forms of psychopathology including anxiety and depressive disorders [16]. Not unexpectedly, then, we hypothesized that these failures would considerably impinge on the reaction to the loss among bereaved caregivers in the context of palliative care. It is worth noting that our findings depicted that people with lower certainty about the mental states of the self and others had higher levels of anxiety. Together with gender, this model explained nearly a quarter of the variance for anxiety, as evaluated by the HADS-A, regardless of years from the loss and the condition of being the main caregiver. By way of explanation of our finding, certainty about the mental states regards different characteristics of reflective functioning from the uncertainty about mental states [23]. Unlike the latter, certainty about mental states, as measured by the RFQ, tends to be inversely related to a broad range of maladaptive indices, such as the use of primitive defences, trait anger, and interpersonal problems. Conversely, it is positively, although weakly, associated with anger control and well-being.

In this perspective, certainty about the mental states of oneself and others was inversely related (though slightly less) to the symptoms of depression reported by the bereaved. When considering this regression model, our findings also showed that the percentage of explained variance for depression, as measured by the HADS-D, was lesser than for anxiety, even though the role of gender was similar in both. Whatever the cause, as revealed by our results, the condition of being the main caregiver represents a risk factor for depression but not for anxiety. Nonetheless, there is the temptation to conclude that the severe lack of certainty about mental states may have been a predictor of the depression and anxiety experienced by family caregivers. Despite the relationships between mentalizing and reaction to the loss among bereaved individuals remaining underresearched, research in other fields has linked hypermentalizing to borderline personality features [27]. Moreover, difficulties in emotion regulation seem to mediate the tendency to overinterpret mental states or overattribute intentions, indicating hypermentalizing and borderline personality features. Conversely, accordance to any extent implies good mentalizing. Put in the wording of mentalization theory, genuine mentalizing is marked by a certain recognition of the opacity of mental states [12]. Another caveat concerns our findings on the uncertainty about mental states. The fact that it was not a significant predictor of depression and anxiety underpins the view that uncertainty ought to be considered as more able to differentiate between clinical and non-clinical samples [27]. What is noteworthy here is that the participants in this study had no diagnosis of mental disorder insofar as they only struggled with the consequences of the death of a loved one.

Thus, taken together, our findings may further hone the understanding of the role of mentalizing in psychopathology as well as the factors underneath the onset of mental disorders among bereaved individuals. In this vein, our findings may help to disclose that failures of reflective functioning affect the reaction to the loss in the context of palliative care in terms of anxiety and depression. Although more research is still needed, our findings have relevant implications for clinical practice and especially for prevention.

Despite the finding of this study contributing to understanding the onset of mental disorders among family caregivers, corroborating the idea that mentalization-based interventions might be useful for preventing anxiety and depression, some limitations should be taken into account. First, the use of a cross-sectional design did not allow us to gather definitive evidence for a causal relationship between reflective functioning, anxiety, and depression. Thus, future studies should employ a longitudinal design to better clarify this kind of relationship. Second, distortions caused by the oversampling of certain characteristics may not allow us to generalize our results to other bereaved individuals. However, some characteristics such as the female gender can be deemed as emblematic of family caregivers who assisted their loved ones. Indeed, the burgeoning recent literature on informal caregiving has highlighted that this role is mostly assumed by females in a wide array of conditions including advanced cancer [28,29,30,31]. Arguably, the findings of these studies may help to explain the reasons why females tend to suffer from higher levels of depression and other psychological symptoms than men. Third, a motley array of factors may potentially influence anxiety and depression in bereaved individuals, such as the history of the disease. Forthcoming research should take into account their role in predicting anxiety and depression levels after the loss.

## 5. Conclusions

In sum, our results indicate that failures in mentalizing, operationalized as reflective functioning, may increase the risk of symptoms of a dysfunctional reaction to loss. Specifically, an impairment in reflective functioning characterized by a low certainty about the mental states of the self and others was related to more symptoms of anxiety and depression reported by family caregivers of palliative care patients. The most relevant implication for clinical practice of specialists within the field regards the prevention of anxiety and depression symptoms and the promotion of psychological health among bereaved caregivers in the context of palliative home care. Though these findings need further investigation, they support the claim that the lack of certainty about mental states could be underneath the psychological symptoms reported by bereaved individuals. Thus, an assessment of reflective functioning is paramount, while a group psychological intervention [32] to enhance it could be very helpful for decreasing the incidence of anxiety and depression disorders.

## Figures and Tables

**Table 1 ijerph-19-11930-t001:** Demographic and loss characteristics of the sample.

Characteristics	n (%)	M	SD
Age (in years)		43.50	14.04
Gender			
Male	36 (22.9)		
Female	121 (77.1)		
Education			
Primary or middle school diploma	34 (21.7)		
High school diploma	66 (42)		
Graduate	57 (36.3)		
Work after loss			
Yes	94 (59.87)		
No	63 (40.13)		
Relation with the deceased loved one			
Son or daughter	82 (52.3)		
Nephew	32 (20.4)		
Spouse	15 (9.5)		
Other (for example, brother-in-law)	28 (17.8)		
Main caregiver			
Yes	90 (57.3)		
No	67 (42.7)		
Time since the loss (years)		3.59	4.92

Notes: n = 157.

**Table 2 ijerph-19-11930-t002:** Descriptive and correlational analyses.

Variable	Min	Max	M	SD	1. Time since the Loss (Years)	2. HADS-A	3. HADS-D	4. RFQ U
1. Time since the loss (years)	1	28	3.59	4.92				
2. HADS-A	0	19	7.34	4.11	0.04			
3. HADS-D	0	18	6.22	3.56	−0.16 *	0.61 **		
4. RFQ U	0	2.50	0.95	0.53	0.08	0.17 *	0.11	
5. RFQ C	0	2.50	0.90	0.70	−0.11	−0.30 **	−0.21 **	−0.32 **

Notes: n = 157. Abbreviations: Min = minimum value; Max = maximum value; M = mean; SD = standard deviation; HADS-A = HADS-Anxiety; HADS-D = HADS-Depression; RFQ U = RFQ Uncertainty about mental states; RFQ C = RFQ Certainty about mental states. * *p* < 0.05; ** *p* < 0.01.

**Table 3 ijerph-19-11930-t003:** The regression results of the effects of reflective functioning on anxiety.

Predictor of Anxiety	*B*	*b* 95% CI [LL, UL]	β	*sr* ^2^	*R*	Fit	Difference
(Intercept)	0.84 **	[0.53, 1.15]					
Age	−0.01	[−0.01, 0.01]	−0.04	−0.04	−0.04		
Gender	0.44 **	[0.25, 0.64]	0.34	0.34	0.34		
						*R*^2^ = 0.115 **	
(Intercept)	0.86 **	[0.34, 1.38]					
Age	−0.01	[−0.01, 0.01]	−0.05	−0.04	−0.04		
Gender	0.44 **	[0.24, 0.65]	0.34	0.33	0.34		
Years from the loss	−0.01	[−0.02, 0.02]	−0.14	−0.01	0.01		
Main caregiver	−0.01	[−0.21, 0.19]	−0.01	−0.01	−0.03		
						*R*^2^ = 0.115 **	Δ*R*^2^ = 0
(Intercept)	0.89 **	[0.38, 1.40]					
Age	−8.97	[−0.01, 0.01]	−0.01	−0.01	−0.04		
Gender	0.48 **	[0.29, 0.66]	0.36	0.38	0.34		
Years from the loss	−0.01	[−0.02, 0.01]	−0.04	−0.05	0.01		
Main caregiver	−0.02	[−0.21, 0.17]	−0.02	−0.02	−0.03		
RFQ Uncertainty	0.12	[−0.04, 0.28]	0.11	0.12	0.17		
RFQ Certainty	−0.23 **	[−0.35, −0.11]	−0.29	−0.29	−0.30		
						*R*^2^ = 0.228 **	Δ*R*^2^ = 0.112

Notes: n = 157. A significant *b*-weight indicates the beta-weight and semi-partial correlations are also significant. *sr*^2^ represents the semi-partial correlation squared. *r* represents the zero-order correlation. *LL* and *UL* indicate the lower and upper limits of the confidence interval for *B*. ** *p* < 0.01.

**Table 4 ijerph-19-11930-t004:** The regression results of the effects of reflective functioning on depression.

Predictor of Depression	*B*	*b* 95% CI [LL, UL]	β	*sr* ^2^	*R*	Fit	Difference
(Intercept)	0.59 **	[0.29, 0.89]					
Age	0.01	[−0.01, 0.01]	0.05	0.05	0.05		
Gender	0.20 *	[0.01, 0.39]	0.17	0.17	0.17		
						*R*^2^ = 0.030	
(Intercept)	1.12 **	[0.63, 1.62]					
Age	−0.01	[−0.01, 0.01]	−0.06	−0.05	0.05		
Gender	0.18	[−0.01, 0.37]	0.15	0.15	0.17		
Years from the loss	−0.01	[−0.03, 0.01]	−0.07	−0.07	−0.12		
Main caregiver	−0.22 *	[−0.41, −0.04]	−0.21	−0.19	−0.23		
						*R*^2^ = 0.078 *	Δ*R*^2^ = 0.048
(Intercept)	1.16 **	[0.66, 1.66]					
Age	−0.01	[−0.01, 0.01]	−0.02	−0.02	0.05		
Gender	0.20 *	[0.01, 0.39]	0.16	0.17	0.17		
Years from the loss	−0.01	[−0.03, 0.01]	−0.09	−0.09	−0.12		
Main caregiver	−0.23 *	[−0.41, −0.05]	−0.22	−0.20	−0.23		
RFQ Uncertainty	0.07	[−0.09, 0.22]	0.08	0.07	0.11		
RFQ Certainty	−0.17 **	[−0.29, −0.05]	0.06	−0.22	−0.21		
						*R*^2^ = 0.144 **	Δ*R*^2^ = 0.066

Notes: n = 157. A significant *b*-weight indicates the beta-weight and semi-partial correlations are also significant. *sr*^2^ represents the semi-partial correlation squared. *r* represents the zero-order correlation. *LL* and *UL* indicate the lower and upper limits of the confidence interval for *B*. * *p* < 0.05; ** *p* < 0.01.

## Data Availability

The raw data supporting the conclusions of this article will be made available on request by the corresponding author, without undue reservation.

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
