# Peer review of "Failures in Reflective Functioning and Reported Symptoms of Anxiety and Depression in Bereaved Individuals: A Study on a Sample of Family Caregivers of Palliative Care Patients"

_ijerph, 2022, doi:10.3390/ijerph191911930_

Round 1
Reviewer 1 Report
The authors aim to understand how mentalizing impairments affect anxiety and depression, and how reflective functioning influences anxiety and depression. Although the research questions are interesting, the manuscript must be improved before being considered for publication. First, I would appreciate the authors clearly defining what they mean by "mentalizing impairment" and how the concept was operationalized, which is very obscure in the measurement session. How does reflective functioning play a role in the influence of mentalizing impairments on anxiety/depression? Does it moderate the relationship between mentalizing impairment and anxiety/depression? Currently, the manuscript is written as if mentalizing impairment and reflective functioning are two separate independent variables affecting anxiety and depression. The authors should address the relationship between mentalizing impairment and reflective functioning.
Second, more details about the sample are needed. For example, the authors mentioned that the sample was drawn from a larger research project. Was the sample a proportion of the participants involved in the larger project? If not, why? If so, a project with 157 participants does not seem to be "large". More demographic information is needed, such as education, marriage status, etc. The reporting of descriptive and inferential statistics is problematic. For example, the HADS-A scale adopts a 0-3 Likert point scale, but the mean score was above 7. This is very confusing. The authors need to check the measurement and correct the reporting (and possibly re-run the analysis).
Author Response
We are grateful to the reviewer for the insightful comments on the paper. We have been able to incorporate changes to reflect the reviewer’ suggestions, and we substantially revised the manuscript. We have highlighted the changes within the manuscript using track changes.

Reviewer 2 Report
Thanks for this interesting piece of work with a population that is difficult to reach. I like the fact that you describe your analyses clearly, which is not always the case in this field of research.
I do have some remarks and questions for clarification
Small remarks
· I see you oftentimes use the word ‘effect’ (182, 192,215, 25, ….). This is misleading in my opinion as you have a cross-sectional study, which you acknowledge in your limitations. Statistically speaking, it’s correct but I would not use this term in your abstract or conclusions. In other sentences you use the word ‘predict’ which is correct and does not claim things that can only be measured with longitudinal data.
· 113: I would add while “high scores on the RFQ C” indicate hypermentalizing (i.e. over mentalizing disposition….). That would make it more clear as now it seems to be that RCQ C is not explained.
· 20/22: It was not clear to me from the beginning that both elements from the RFQ are actually mentalizing impairments as ‘certainty about mental states’ sounds as if it’s a positive quality of someone. Would it be an option to write something like ‘to be overly certain about mental states’ or ‘hypermentalizing’? I know you make it clear in your title and also explains it later on but oftentimes persons only read your abstract
· 194: ‘decrease’, I did not get why you are writing about ‘decrease’ here?
· 209: ‘certain characteristics’, are you pointing to something else than gender? Which characteristics other than that are oversampled? Do you have a reference to state that females are more often family caregivers?
· 213: ‘imbalance between’, that’s not correct I suppose as you never look into the relation between certainty and uncertainty about mental states?
· I guess that there are also other elements that predict anxiety and depression in bereaved patients. Of course, you cannot control for all of them but you should acknowledge this in your limitation section.
· 77: I would like to know not only the SD for the ‘time since loss’ but also the minimum and maximum.
· 60: two interspacing’s
· 54: is number 9 the correct reference for this? It’s about eating disorders and not about bereaved?
Elements that are in need of clarification
· Why did you use the RFQ? As far as I know it’s not meant to be used for non-psychiatric patients like bereaved individuals and actually should be used in patients with borderline. You did not target for patients with prolonged grief disorder but just for family caregivers whereby most of them probably show no maladaptive reactions (see line 53). You acknowledge that there are not enough studies in the field of bereaved individuals and that’s the reason why you conduct this research. Are there other questionnaires available that see this mentalizing as a transdiagnostic construct as you describe?
· What is the rationale behind your hierarchical regression? It would be helpful if you could explain what the theoretical reasons are to first add age and gender, and only in step 3 uncertainty and certainty.
· I missed your hypotheses about the relationship between ‘years form the loss’ and ‘condition of being the main caregiver’. I guess you could have expected that, when being the main caregiver and having RFQ problems, higher rates of anxiety/depression would be seen in comparison to those that were not the main caregiver. What’s your hypothesis about ‘years from the loss’? If it was just as background variable and no specific hypotheses were available already, you could mention that.
· I missed your ideas about potential explanations why certainty and not uncertainty predicted anxiety and depression. I would add that to the discussion.
Author Response

(The authors gave the same response as above.)

Round 2
Reviewer 1 Report
The manuscript has been significantly improved and clarified. I appreciate the authors' work. I wonder whether uncertainty and certainty of RFQ are just the flipsides of the same coin. Some example items for the measure can be helpful. Inclusion of both subscales can introduce additional error terms into the regression model and attenuate the effects.
Author Response
We would like to thank the Reviewer so much for carefully reading our manuscript and the comments she/he raised.
Please, find attached the detailed replies.

Reviewer 2 Report
Thanks for taking up the earlier remarks, it made the paper already much better and understandable. I still have a few remarks:
- the biggest problem I have right now with the paper, is the fact that the authors mix up 'depression and anxiety symptoms' and 'disorder'. Sometimes, they talk about the former, sometimes about the latter. This is confusing and also shows that your theoretical framework and rationale behind the research is not on point yet. On the one hand, you describe that only a small amount of bereaved individuals show really maladaptive reactions and they do not show psychopathology but rather very normal depressive symptoms and anxiety symptoms that maybe will fade throughout time. On the other hand, you wrote some new parts about psychopathology and the link with mentalizing. Make sure that your research question is clear. Personally, it seems to me like you did not target to look into bereaved individuals with psychopathology, so I do not get why you focus sometimes on complicated grief and depression and anxiety disorders. I think (correct me if I'm wrong) that the HADS also does not claim to be able to detect major disorders but just gives a glimp of the symptoms of depression, normally even especially for inpatients that also suffer from physical pain. The HADS takes into account not to mix up physical pain from an illness and depressive/anxiety symptoms. The HADS is not based on the DSM, what would be used to really see it as a disorder. Please revise your theoretical framework and make sure the research question is very clear.
- small remark: I missed the number of participants that you asked to participate and the reasons why some would not consent. You do not describe correctly how you sampled the participants.
- measures: I would like to have an example of an item of all the subscales of the two questionnaires.
- sentence 119-121 seem to repeat each other
- line 126: I did not get 'not supported by the evidence'. What evidence?
- line 139: it seems to me like you mixed up first and second model here
- table 4: 'predictor of anxiety' in the wrong place
- line 222: the two subscales share four items? How is that possible as they measure opposite things? I would make sure that this is explained under 'measures' and not just mentioned on page 7.
- abstract: if allowed by the journal, you can divide the abstract in method, results section (or more sections). Now, you use 'methods:' but no other subdivisions, which makes it strange to read.
Author Response

(The authors gave the same response as above.)
